# Constraints Based Convex Belief Propagation

**Yaniv Tenzer**
Department of Statistics
The Hebrew University

**Alexander Schwing**
Department of Electrical and Computer Engineering
University of Illinois at Urbana-Champaign

**Kevin Gimpel**
Toyota Technological Institute at Chicago

**Tamir Hazan**
Faculty of Industrial Engineering and Management
Technion - Israel Institute of Technology

## Abstract

Inference in Markov random fields subject to consistency structure is a fundamental problem that arises in many real-life applications. In order to enforce consistency, classical approaches utilize consistency potentials or encode constraints over feasible instances. Unfortunately this comes at the price of a tremendous computational burden. In this paper we suggest to tackle consistency by incorporating constraints on beliefs. This permits derivation of a closed-form message-passing algorithm which we refer to as the Constraints Based Convex Belief Propagation (CBCBP). Experiments show that CBCBP outperforms the conventional consistency potential based approach, while being at least an order of magnitude faster.

## 1 Introduction

Markov random fields (MRFs) [10] are widely used across different domains from computer vision and natural language processing to computational biology, because they are a general tool to describe distributions that involve multiple variables. The dependencies between variables are conveniently encoded via potentials that define the structure of a graph.

Besides encoding dependencies, in a variety of real-world applications we often want consistent solutions that are physically plausible, *e.g.*, when jointly reasoning about multiple tasks or when enforcing geometric constraints in 3D indoor scene understanding applications [18]. Therefore, various methods [22, 13, 16, 12, 1] enforce consistency structure during inference by imposing constraints on the feasible instances. This was shown to be effective in practice. However for each new constraint we may need to design a specifically tailored algorithm. Therefore, the most common approach to impose consistency is usage of PN-consistency potentials [9]. This permits reuse of existing message passing solvers, however, at the expense of an additional computational burden, as real-world applications may involve hundreds of additional factors.

Our goal in this work is to bypass this computational burden while being generally applicable. To do so, we consider the problem of inference when probabilistic equalities are imposed over the beliefs of the model rather than its feasible instances. As we show in Sec. 3, the adaptive nature of message passing algorithms conveniently allows for such probabilistic equality constraints within its framework. Since our method eliminates potentially many multivariate factors, inference is much more scalable than using PN-consistency potentials [9].

In this paper, for notational simplicity, we illustrate the belief constraints based message passing rules using a framework known as convex belief propagation (CBP). We refer to the illustrated algorithm as constraints based CBP (CBCBP). However we note that the same derivation can be used to obtain, *e.g.*, a constraints based tree-reweighted message passing algorithm.

We evaluate the benefits of our algorithm on semantic image segmentation and machine translation tasks. Our results indicate that CBCBP improves accuracy while being at least an order of magnitude faster than CBP.

## 2  Background

In this section we review the standard CBP algorithm. To this end we consider joint distributions defined over a set of discrete random variables $X = (X_1, \ldots, X_n)$. The distribution $p(x_1, \ldots, x_n)$ is assumed to factor into a product of non-negative potential functions, *i.e.*, $p(x_1, \ldots, x_n) \propto \exp\left(\sum_r \theta_r(x_r)\right)$, where $r \subset \{1, \ldots, n\}$ is a subset of variable indices, which we use to restrict the domain via $x_r = (x_i)_{i \in r}$. The real-valued functions $\theta_r(x_r)$ assign a preference to each of the variables in the subset $r$. To visualize the factorization structure we use a region graph, *i.e.*, a generalization of factor graphs. In this graph, each real-valued function $\theta_r(x_r)$ corresponds to a node. Nodes $\theta_r$ and $\theta_p$ can be connected if $r \subset p$. Hence the parent set $P(r)$ of a region $r$ contains index sets $p \in P(r)$ if $r \subset p$. Conversely we define the set of children of region $r$ as $C(r) = \{c : r \in P(c)\}$.

An important inference task is computation of the marginal probabilities $p(x_r) = \sum_{x \setminus x_r} p(x)$. Whenever the region graph has no cycles, marginals are easily computed using belief propagation. Unfortunately, this algorithm may not converge in the presence of cycles. To fix convergence a variety of approximations have been suggested, one of which is known as convex belief propagation (CBP).

CBP performs block-coordinate descent over the dual function of the following program:

$$\max_{b_r} \sum_{r, x_r} b_r(x_r)\theta_r(x_r) + \sum_r H(b_r) \text{ s.t. } \begin{cases} \forall r & b_r(x_r) \geq 0, \sum_{x_r} b_r(x_r) = 1, \\ \forall r, p \in P(r), x_r & \sum_{x_p \setminus x_r} b_p(x_p) = b_r(x_r). \end{cases} \quad (1)$$

This program is defined over marginal distributions $b_r(x_r)$ and incorporates their entropy $H(b_r)$ in addition to the potential function $\theta_r$.

In many real world applications we require the solution to be consistent, *i.e.*, hard constraints between some of the involved variables exist. For example, consider the case where $X_1, X_2$ are two binary variables such that for every feasible joint assignment, $x_1 = x_2$. To encourage consistency while reusing general purpose solvers, a PN-consistency potential [9] is often incorporated into the model:

$$\theta_{1,2}(x_1, x_2) = \begin{cases} 0 & x_1 = x_2 \\ -c & \text{otherwise} \end{cases} . \quad (2)$$

Hereby $c$ is a positive constant that is tuned to penalize for the violation of consistency. As c increases, the following constraint holds:

$$b_1(X_1 = x_1) = b_2(X_2 = x_2). \quad (3)$$

However, usage of PN-potentials raises concerns: (i) increasing the number of pairwise constraints decreases computational efficiency, (ii) enforcing consistency in a soft manner requires tuning of an additional parameter $c$, (iii) large values of $c$ reduce convergence, and (iv) large values of $c$ result in corresponding beliefs being assigned zero probability mass which is not desirable.

To alleviate these issues we suggest to enforce the equality constraints given in Eq. (3) directly during optimization of the program given in Eq. (1). We refer to the additionally introduced constraints as *consistency constraints*.

At this point two notes are in place. First we emphasize that utilizing consistency constraints instead of PN-consistency potentials has a computational advantage, since it omits all pairwise beliefs that correspond to consistency potentials. Therefore it results in an optimization problem with fewer functions, which is expect to be more efficiently solvable.

Second we highlight that the two approaches are not equivalent. Intuitively, as c increases, we expect consistency constraints to yield better results than usage of PN-potentials. Indeed, as c increases, the PN-consistency potential enforces the joint distribution to be diagonal, *i.e.*, $b(X_1 = i, X_2 = j) = 0$, $\forall i \neq j$. However, the consistency constraint as specified in Eq. (3) only requires the univariate marginals to agree. The latter is a considerably weaker requirement, as a diagonal pairwise distribution implies agreement of the univariate marginals, but the opposite direction does not hold. Consequently, using consistency constraints results in a larger search space, which is desirable.

**Algorithm 1** Constraints Based Convex Belief Propagation (CBCBP)

---

Repeat until convergence:
Update $\lambda$ messages - for each $r$ update for all $p \in P(r), x_r$:

$$\mu_{p \to r}(x_r) = \ln \sum_{x_p \backslash x_r} \exp \left( \theta_r(x_r) - \sum_{p' \in P(p)} \lambda_{p \to p'}(x_p) + \sum_{r' \in C(p) \backslash r} \lambda_{r' \to p}(x_{r'}) - \sum_{k \in K_p} \nu_{p \to k}(x_p) \right)$$

$$\lambda_{r \to p}(x_r) \propto \frac{1}{1 + |P(r)|} \left( \theta_r(x_r) + \sum_{c \in C(r)} \lambda_{c \to r}(x_c) + \sum_{p \in P(r)} \mu_{p \to r}(x_r) - \sum_{k \in K_r} \nu_{r \to k}(x_r) \right) - \mu_{p \to r}(x_r)$$

Update $\nu$ messages - for each $k \in K$ update for all $r \in N(k)$ using $\alpha_{r,k}$ as defined in Eq. (6):

$$\nu_{r \to k}(s_r^k) = \log \alpha_{r,k} - \frac{1}{|N(k)|} \sum_{r' \in N(k)} \log \alpha_{r',k}$$

---

Figure 1: The CBCBP algorithm. Shown are the update rules for the $\lambda$ and $\nu$ messages.

Next we derive a general message-passing algorithm that aims at solving the optimization problem given in Eq. (1) subject to consistency constraints of the form given in Eq. (3).

## 3  Constraints Based Convex Belief Propagation (CBCBP)

To enforce consistency of beliefs we want to incorporate constraints of the form $b_{r_1}(x_{r_1}) = \ldots = b_{r_m}(x_{r_m})$. Each constraint involves a set of regions $r_i$ and some of their assignments $x_{r_i}$. If this constraint involves more than two regions, *i.e.*, if $m > 2$, it is easier to formulate the constraint as a series of constraints $b_{r_i}(x_{r_i}) = v$, $i \in \{1, \ldots, m\}$, for some constant $v$ that eventually cancels.

Generally, given a constraint $k$, we define the set of its neighbours $N(k)$ to be the involved regions $r_i^k$ as well as the involved assignment $x_{r_i}^k$, *i.e.*, $N(k) = \{r_i^k, x_{r_i}^k\}_{i=1}^{m_k}$. To simplify notation we subsequently use $r \in N(k)$ instead of $(r, x_r) \in N(k)$. However, it should be clear from the context that each region $r^k$ is matched with a value $x_r^k$.

We subsume all constraints within the set $K$. Additionally, we let $K_r$ denote the set of all those constraints $k$ which depend on region $r$, *i.e.*, $K_r = \{k : r \in N(k)\}$.

Using the aforementioned notation we are now ready to augment the conventional CBP given in Eq. (1) with one additional set of constraints. The CBCBP program then reads as follows:

$$\max_{b_r} \sum_{r, x_r} b_r(x_r) \theta_r(x_r) + \sum_r H(b_r) \quad \text{s.t.} \quad \begin{cases} \forall r & b_r(x_r) \geq 0, \sum_{x_r} b_r(x_r) = 1 \\ \forall r, p \in P(r), x_r & \sum_{x_p \backslash x_r} b_p(x_p) = b_r(x_r) \\ \forall k \in K, r \in N(k) & b_r(x_r^k) = v_k \end{cases} .$$

$$(4)$$

To solve this program we observe that its constraint space exhibits a rich structure, defined on the one hand by the parent set $P$, and on the other hand by the neighborhood of the constraint subsumed in the set $K$. To exploit this structure, we aim at deriving the dual which is possible because the program is strictly convex. Importantly we can subsequently derive block-coordinate updates for the dual variables, which are efficiently computable in closed form. Hence solving the program given in Eq. (4) via its dual is much more effective. In the following we first present the dual before discussing how to efficiently solve it.

**Derivation of the dual program:**   The dual program of the task given in Eq. (4) is obtained by using the Lagrangian as shown in the following lemma.

**Lemma 3.1.:** *The dual problem associated with the primal program given in Eq.* (4) *is:*

$$\min_{\lambda, \nu} \sum_r \log \sum_{x_r} \exp \left( \theta_r(x_r, \lambda) - \sum_{k \in K_r} \nu_{r \to k}(x_r) \right) \quad s.t. \quad \forall k \in K, \sum_{r \in N(k)} \nu_{r \to k}(x_r^k) = 0,$$

*where we set* $\nu_{r\to k}(x_r) = 0 \ \forall k \in K, r \in N(k), x_r \neq x_r^k$ *and where we introduced* $\theta_r(x_r, \lambda) = \theta_r(x_r) - \sum_{p\in P(r)} \lambda_{r\to p}(x_r) + \sum_{c\in C(r)} \lambda_{c\to r}(s_c).$

**Proof:** We begin by defining a Lagrange multiplier for each of the constraints given in Eq. (4). Concretely, for all $r, p \in P(r), x_r$ we let $\lambda_{r\to p}(x_r)$ be the Lagrange multiplier associated with the marginalization constraint $\sum_{x_p\backslash x_r} b_p(x_p) = b_r(x_r)$. Similarly for all $k \in K, r \in N(k)$, we let $\nu_{r\to k}(x_r^k)$ be the Lagrange multiplier that is associated with the constraint $b_r(x_r^k) = v_k$.

The corresponding Lagrangian is then given by

$$L(b, \lambda, \nu) = \sum_{r, x_r} b_r(x_r) \left( \theta_r(x_r, \lambda) - \sum_{k\in K_r} \nu_{r\to k}(x_r) \right) + \sum_r H(b_r) + \sum_{k\in K, r\in N(k)} \nu_{r\to k}(x_r^k) v_k,$$

where $\theta_r(x_r, \lambda) = \theta_r(x_r) - \sum_{p\in P(r)} \lambda_{r\to p}(x_r) + \sum_{c\in C(r)} \lambda_{c\to r}(x_c)$ and $\nu_{r\to k}(x_r) = 0$ for all $k, r \in N(k), x_r \neq x_r^k$.

Due to conjugate duality between the entropy and the log-sum-exp function [25], the dual function is:

$$D(\lambda, \nu) = \max_b L(b, \lambda, \nu) = \sum_r \log \sum_{x_r} \exp\left( \theta_r(x_r, \lambda) - \sum_{k\in K_r} \nu_{r\to k}(x_r) \right) + \sum_k v_k \sum_{r\in N(k)} \nu_{r\to k}(x_r^k).$$

The result follows since the dual function is unbounded from below with respect to the Lagrange multipliers $\nu_{r\to k}(x_r^k)$, requiring constraints. ∎

**Derivation of message passing update rules:** As mentioned before we can derive block-coordinate descent update rules for the dual which are computable in closed form. Hence the dual given in Lemma 3.1 can be solved efficiently, which is summarized in the following theorem:

**Theorem 3.2.:** A block coordinates descent over the dual problem giving in Lemma 3.1 results in a message passing algorithm whose details are given in Fig. 1 and which we refer to as the CBCBP algorithm. It is guaranteed to converge.

Before proving this result, we provide intuition for the update rules: as in the standard and distributed [19] CBP algorithm, each region $r$ sends a message to its parents via the dual variable $\lambda_{r\to p}$. Differently from CBP but similar to distributed variants [19], our algorithm has another type of messages, *i.e.*, the $\nu$ messages. Conceptually, think of the constraints as a new node. A constraint node $k$ is connected to a region $r$ if $r \in N(k)$. Hence, a region $r$ 'informs' the constraint node using the dual variable $\nu_{r\to k}$. We now show how to derive the message passing rules to optimize the dual.

**Proof:** First we note that convergence is guaranteed by the strict convexity of the primal problem [6].

Next we begin by optimizing the dual function given in Lemma 3.1 with respect to the $\lambda$ parameters. Specifically, for a chosen region $r$ we optimize the dual w.r.t. a block of Lagrange multipliers $\lambda_{r\to p}(x_r) \ \forall p \in P(r), x_r$. To this end we derive the dual with respect to $\lambda_{r\to p}(x_r)$ while keeping all other variables fixed. The technique for solving the optimality conditions follows existing literature, augmented by messages $\nu_{r\to k}$. It yields the update rules given in Fig. 1.

Next we turn to optimizing the dual with respect to the Lagrange multipliers $\nu$. Recall that each constraint $k \in K$ in the dual function given in Lemma 3.1 is associated with the linear constraint $\sum_{r\in N(k)} \nu_{r\to k}(x_r^k) = 0$. Therefore we employ a Lagrange multiplier $\gamma_k$ for each $k$. For compact exposition, we introduce the Lagrangian that is associated with a constraint $k$, denoted by $L_k$:

$$L_k(\lambda, \nu) = \sum_{r\in N(k)} \log \sum_{x_r} \exp\left( \theta_r(x_r, \lambda) - \sum_{k\in K_r} \nu_{r\to k}(x_r) \right) + \gamma_k \left( \sum_{r\in N(k)} \nu_{r\to k}(x_r^k) \right).$$

Deriving $L_k$ with respect to $\nu_{r\to k} \ \forall r \in N(k)$ and using optimality conditions, we then arrive at:

$$\nu_{r\to k}(x_r^k) = \log\left( \alpha_{r,k} \cdot \frac{1 + \gamma_k}{-\gamma_k} \right) \tag{5}$$

for all $r \in N(k)$, where

$$\alpha_{r,k} = \frac{\exp\left( \theta_r(x_r^k, \lambda) - \sum_{k'\in K_r\backslash k} \nu_{r\to k'}(x_r^k) \right)}{\sum_{x_r\backslash x_r^k} \exp\left( \theta_r(x_r, \lambda) - \sum_{k'\in K_r} \nu_{r\to k'}(x_r) \right)}. \tag{6}$$

| | $n = 100$ | $n = 200$ | $n = 300$ | $n = 400$ |
|---|---|---|---|---|
| CBP | $1.47 \pm 2e^{-4}$ | $2.7 \pm 1e^{-4}$ | $5.95 \pm 3e^{-3}$ | $13.42 \pm 2e^{-3}$ |
| CBCBP | $0.05 \pm 3e^{-4}$ | $0.11 \pm 1e^{-4}$ | $0.23 \pm 2e^{-3}$ | $0.43 \pm 1e^{-3}$ |

Table 1: Average running time and standard deviation, over 10 repetitions, of CBCBP and CBP. Both infer the parameters of MRFs that consist of $n$ variables.

| | $c = 2$ | $c = 4$ | $c = 6$ | $c = 8$ | $c = 10$ |
|---|---|---|---|---|---|
| CBP | $31.40 \pm 0.74$ | $42.05 \pm 1.02$ | $49.17 \pm 1.27$ | $53.35 \pm 0.93$ | $58.01 \pm 0.82$ |

Table 2: Average speedup factor and standard deviation, over 10 repetitions, of CBCBP compared to CBP, for different values of $c$. Both infer the beliefs of MRFs that consist of 200 variables.

Summing the right hand side of Eq. (5) over $r \in N(k)$ and using the constraint $\sum_{r \in N(k)} \nu_{r \to k}(x_r^k) = 0$ results in

$$\frac{1 + \gamma_k}{-\gamma_k} = \left( \prod_{r \in N(k)} \frac{1}{\alpha_{r,k}} \right)^{\frac{1}{|N(k)|}}.$$

Finally, substituting this result back into Eq. (5) yields the desired update rule. ∎

We summarized the resulting algorithm in Fig. 1 and now turn our attention to its evaluation.

## 4 Experiments

We first demonstrate the applicability of the procedure using synthetic data. We then turn to image segmentation and machine translation, using real-world datasets. As our work directly improves the standard CBP approach, we use it as a baseline.

### 4.1 Synthetic Evaluation

Consider two binary variables $X$ and $Y$ whose support consists of L levels, $\{1, \ldots, L\}$. Assume we are given the following PN-consistency potential:

$$\theta_{x,y}(x, y) = \begin{cases} 0 & (y = 1 \wedge x = 1) \vee (y = 0 \wedge x \neq 1) \\ -c & \text{otherwise,} \end{cases} \tag{7}$$

where $c$ is some positive parameter. This potential encourages the assignment $y = 1$ to agree with the assignment $x = 1$ and $y = 0$ to agree with $x = \{2, \ldots, L\}$. Phrased differently, this potential favours beliefs such that:

$$b_y(y = 1) = b_x(x = 1), \quad b_y(y = 0) = b_x(x \neq 1). \tag{8}$$

Therefore, one may replace the above potential using a single consistency constraint. Note that the above two constraints complement each other, hence, it suffices to include one of them. We use the left consistency constraint since it fits our derivation.

We test this hypothesis by constructing four networks that consist of $n = 2v$, $v = 50, 100, 150, 200$ variables, where $v$ variables are binary, denoted by $\mathbf{Y}$ and the other $v$ variables are multi-levels, subsumed within $\mathbf{X}$. Note that the support of variable $X_i$, $1 \leq i \leq v$, consists of $i$ states. Each multi-level variable is matched with a binary one. For each variable we randomly generate unary potentials according to the standard Gaussian distribution.

We then run the standard CBP algorithm using the aforementioned PN-consistency potential given in Eq. (7) with $c = 1$. In a next step we replace each such potential by its corresponding consistency constraint following Eq. (8). For each network we repeat this process 10 times and report the mean running time and standard deviation in Tab. 1.

As expected, CBCBP is significantly faster than the standard CBP. Quantitatively, CBCBP was approximately 25 times faster for the smallest, and more than 31 times faster for the largest graphs.

Obviously, different values of $c$ effect the convexity of the problem and therefore also the running time of both CBP and CBCBP. To quantify its impact we repeat the experiment with $n = 200$ for distinct values of $c \in \{2, 4, 6, 8, 10\}$. In Tab. 2 we report the mean speedup factor over 10 repetitions, for each value of $c$. As clearly evident, the speedup factors substantially increases with $c$.

|  | global accuracy | average accuracy | mean running time |
|---|---|---|---|
| CBP | 84.2 | 74.3 | $1.41 \pm 5e^{-3}$ |
| CBCBP | 85.4 | 76.1 | $0.02 \pm 2e^{-3}$ |

Table 3: Global accuracy, average accuracy and mean running time as well as standard deviation for the 256 images of the MSRC-21 dataset.

|  | void | grass | tree | cow | sheep | sky | aeropl | water | face | car | bicycle | flower | sign | bird | book | chair | road | cat | dog | body | boat |
|---|---|---|---|---|---|---|---|---|---|---|---|---|---|---|---|---|---|---|---|---|---|
| CBP | **0.79** | **0.99** | 0.84 | 0.68 | 0.67 | 0.92 | 0.78 | **0.83** | **0.82** | 0.79 | 0.90 | 0.92 | 0.56 | 0.42 | 0.94 | 0.48 | 0.87 | **0.81** | 0.51 | 0.63 | **0** |
| CBCBP | 0.72 | 0.97 | **0.89** | **0.77** | **0.84** | **0.95** | **0.83** | **0.83** | **0.82** | **0.80** | **0.92** | **0.96** | **0.69** | **0.49** | **0.95** | **0.58** | **0.89** | **0.81** | **0.53** | **0.65** | **0** |

Table 4: Segmentation accuracy per class of CBCBP and CBP, for the MSRC-21 dataset.

## 4.2 Image Segmentation

We evaluate our approach on the task of semantic segmentation using the MSRC-21 dataset [21] as well as the PascalVOC 2012 [4] dataset. Both contain 21 foreground classes. Each variable $X_i$ in our model corresponds to a super-pixel in an image. In addition, each super-pixel is associated with a binary variable $Y_i$, that indicates whether the super-pixel belongs to the foreground, *i.e.*, $y_i = 1$, or to the background, *i.e.*, $y_i = 0$. The model potentials are:

**Super-pixel unary potentials:** For MSRC-21 these potentials are computed by averaging the TextonBoost [11] pixel-potentials inside each super-pixel. For the PascalVOC 2012 dataset we train a convolutional neural network following the VGG16 architecture.

**Foreground/Background unary potentials:** For MSRC-21 we let the value of the potential at $y_i = 1$ equal the value of the super-pixel unary potential that corresponds to the 'void' label, and for $y_i = 0$ we define it to be the maximum value of the super-pixel unary potential among the other labels. For PascalVOC 2012 we obtain the foreground/background potential by training another convolutional neural network following again the VGG16 architecture.

**Super pixel - foreground/background consistency:** We define pairwise potentials between super-pixel and the foreground/background labels using Eq. (7) and set $c = 1$.

Naturally, these consistency potentials encourage CBP to favour beliefs where pixels that are labeled as 'void' are also labeled as 'background' and vice versa. This can also be formulated using the constraints $b_i(X_i = 0) = b_i(Y_i = 0)$ and $b_i(X_i \neq 1) = b_i(Y_i = 1)$.

We compare the CBCBP algorithm with the standard CBP approach. For MSRC-21 we use the standard error measure of average per class accuracy and average per pixel accuracy, denoted as *global*. Performances results are provided in Tab. 3.

Appealingly, our results indicate that CBCBP outperforms the standard CBP, across both metrics. Moreover and as summarized in Tab. 4, in 19 out of 21 classes CBCBP achieves an accuracy that is equal to or higher than CBP. At last, CBCBP is more than 65 times faster than CBP.

In Tab. 5 we present the average pixel accuracy as well as the Intersection over Union (IoU) metric for the VOC2012 data. We observe CBCBP to perform better since it is able to transfer information between the foreground-background classification and the semantic segmentation.

## 4.3 Machine Translation

We now consider the task of machine translation. We define a phrase-based translation model as a factor graph with many large constraints and use CBCBP to efficiently incorporate them during inference. Our model is inspired by the widely-used approach of [8]. Given a sentence in a source language, the output of our phrase-based model consists of a segmentation of the source sentence into phrases (subsequences of words), a phrase translation for each source phrase, and an ordering of the phrase translations. See Fig. 2 for an illustration.

We index variables in our model by $i = 1, \ldots, n$, which include source words ($sw$), source phrases ($sp$), and translation phrase slots ($tp$). The sequence of source words is first segmented into source phrases. The possible values for source word $sw$ are $X_{sw} = \{(sw_1, sw_2) : (sw_1 \leq sw \leq sw_2) \land (sw_2 - sw_1 < m)\}$, where $m$ is the maximum phrase length.

If source phrase $sp$ is used in the derivation, we say that $sp$ aligns to a translation phrase slot $tp$. If $sp$ is not used, it aligns to $\emptyset$. We define variables $X_{sp}$ to indicate what $sp$ aligns to: $X_{sp} = \{tp :$

| | average accuracy | IOU |
|---|---|---|
| CBP | 90.6 | 62.69 |
| CBCBP | 91.6 | 62.97 |

Table 5: Average accuracy and IOU accuracy for the 1449 images of the VOC2012 dataset.

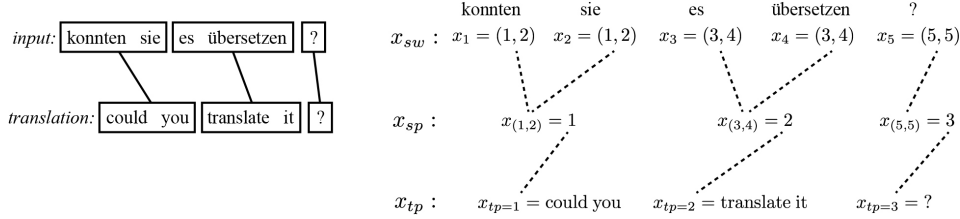

Figure 2: An example German sentence with a derivation of its translation. On the right we show the $x_{sw}$ variables, which assign source words to source phrases, the $x_{sp}$ variables, which assign source phrases to translation phrase slots, and the $x_{tp}$ variables, which fill slots with actual words in the translation. Dotted lines highlight how $x_{sw}$ values correspond to indices of $x_{sp}$ variables, and $x_{sp}$ values correspond to indices of $x_{tp}$ variables. The $x_{sp}$ variables for unused source spans (*e.g.*, $x_{(1,1)}$, $x_{(2,4)}$, *etc.*) are not shown.

$sw_1 - d \leq tp \leq sw_2 + d\} \cup \{\emptyset\}$, *i.e.*, all translation phrase slots $tp$ (numbered from left to right in the translation) such that the slot number is at most distance $d$ from an edge of $sp$.[1]

Each translation phrase slot $tp$ generates actual target-language words which comprise the translation. We define variables $X_{tp}$ ranging over the possible target-language word sequences (**translation phrases**) that can be generated at slot $tp$. However, not all translation phrase slots must be filled in with translations. Beyond some value of $tp$ (equaling the number of source phrases used in the derivation), they must all be empty. To enforce this, we also permit a null ($\emptyset$) translation.

**Consistency constraints:** Many derivations defined by the discrete product space $X_1 \times \cdots \times X_n$ are semantically inconsistent. For example, a derivation may place the first source word into the source phrase $(1, 2)$ and the second source word into $(2, 3)$. This is problematic because the phrases overlap; each source word must be placed into exactly one source phrase. We introduce source word consistency constraints:

$$\forall sp, \forall sw \in sp : \quad b_{sw}(sp) = b(sp).$$

These constraints force the source word beliefs $b_{sw}(x_{sw})$ to agree on their span. There are other consistencies we wish to enforce in our model. Specifically, we must match a source phrase to a translation phrase slot if and only if the source phrase is consistently chosen by all of its source words. Formally,

$$\forall \ sp : \quad b(sp) = \sum_{x_{sp} \neq \emptyset} b_{sp}(x_{sp}).$$

**Phrase translation potentials:** We use pairwise potential functions between source phrases $sp = (sw_1, sw_2)$ and their aligned translation phrase slots $tp$. We include a factor $\langle sp, tp \rangle \in E$ if $sw_1 - d \leq tp \leq sw_2 + d$. Letting $\pi_{sp}$ be the actual words in $sp$, the potentials $\theta_{sp,tp}(x_{sp}, x_{tp})$ determine the preference of the phrase translation $\langle \pi_{sp}, x_{tp} \rangle$ using a phrase table feature function $\tau : \langle \pi, \pi' \rangle \to \mathbb{R}^k$. In particular, $\theta_{sp,tp}(x_{sp}, x_{tp}) = \gamma_p^\top \tau(\langle \pi_{sp}, x_{tp} \rangle)$ if $x_{sp} = tp$ and a large negative value otherwise, where $\gamma_p$ is the weight vector for the Moses phrase table feature vector.

**Language model potentials:** To include $n$-gram language models, we add potentials that score pairs of consecutive target phrases, *i.e.*, $\theta_{tp-1,tp}(x_{tp-1}, x_{tp}) = \gamma_\ell \sum_{i=1}^{|x_{tp}|} \log \Pr(x_{tp}^{(i)} | x_{tp-1} \cdot x_{tp}^{(1)} \cdot \ldots \cdot x_{tp}^{(i-1)})$, where $|x_{tp}|$ is the number of words in $x_{tp}$, $x_{tp}^{(i)}$ is the $i$-th word in $x_{tp}$, $\cdot$ denotes string concatenation, and $\gamma_\ell$ is the feature weight. This potential sums $n$-gram log-probabilities of words in the second of the two target phrases. Internal $n$-gram features and the standard word penalty feature [7] are computed in the $\theta_{tp}$ potentials, since they depend only on the words in $x_{tp}$.

**Source phrase separation potentials:** We use pairwise potentials between source phrases to prevent them aligning to the same translation slot. We also prevent two overlapping source phrases

| | %BLEU |
|---|---|
| no $sw$ constraints | 13.12 |
| $sw$ constraints with CBCBP | 16.73 |

Table 6: %BLEU on test set, showing the contribution of the source word consistency constraints.

from both aligning to non-null slots (*i.e.*, one must align to $\emptyset$). We include a factor between two sources phrases if there is a translation phrase that may relate to both, namely $\langle sp_1, sp_2 \rangle \in E$ if $\exists\, tp :$ $\langle sp_1, tp \rangle \in E$, $\langle sp_2, tp \rangle \in E$. The source phrase separation potential $\theta_{sp_1,sp_2}(x_{sp_1}, x_{sp_2})$ is $-\infty$ if either $x_{sp_1} = x_{sp_2} \neq \emptyset$ or $sp_1 \cap sp_2 \neq \emptyset \wedge x_{sp_1} \neq \emptyset \wedge x_{sp_2} \neq \emptyset$. Otherwise, it is $-\gamma_d |(\delta(sp_1, sp_2) - |x_{sp_1} - x_{sp_2}|)|$, where $\delta(sp_1, sp_2)$ returns the number of source words between the spans $sp_1$ and $sp_2$. This favors similar distances between source phrases and their aligned slots.

**Experimental Setup:** We consider German-to-English translation. As training data for constructing the phrase table, we use the WMT2011 parallel data [2], which contains 1.9M sentence pairs. We use the phrase table to compute $\theta_{sp,tp}$ and to fill $X_{tp}$. We use a bigram language model estimated from the English side of the parallel data along with 601M tokens of randomly-selected sentences from the Linguistic Data Consortium's Gigaword corpus. This is used when computing the $\theta_{tp-1,tp}$ potentials.

As our test set, we use the first 150 sentences from the WMT2009 test set. Results below are (uncased) %BLEU scores [17] on this 150-sentence set.

We use maximum phrase length $m = 3$ and distortion limit $d = 3$. We run 250 iterations of CBCBP for each sentence. For the feature weights ($\gamma$), we use the default weights in Moses, since our features are analogous to theirs. Learning the weights is left to future work.

**Results:** We compare to a simplified version of our model that omits the $sw$ variables and all constraints and terms pertaining to them. This variation still contains all $sp$ and $tp$ variables and their factors. This comparison shows the contribution of our novel handling of consistency constraints. Tab. 6 shows our results. The consistency constraints lead to a large improvement for our model at negligible increase in runtime due to our closed-form update rules. We found it impractical to attempt to obtain these results using the standard CBP algorithm for any source sentences of typical length.

For comparison to a standard benchmark, we also trained a Moses system [7], a state-of-the-art phrase-based system, on the same data. We used default settings and feature weights, except we used max phrase length 3 and no lexicalized reordering model, in order to more closely match the setting of our model. The Moses %BLEU on this dataset is 17.88. When using the source word consistency constraints, we are within 1.2% of Moses. Our model has the virtue of being able to compute marginals for downstream applications and also permits us to study particular forms of constraints in phrase-based translation modeling. Future work can add or remove constraints like we did in our experiments here in order to determine the most effective constraints for phrase-based translation. Our efficient inference framework makes such exploration possible.

## 5    Related Work

Variational approaches to inference have been extensively studied in the past. We address approximate inference using the entropy barrier function and there has been extensive work in this direction, *e.g.*, [24, 14, 23, 5, 19, 20] to name a few. Our work differs since we incorporate consistency constraints within the inference engine. We show that closed-form update rules are still available.

Consistency constraints are implied when using PN-potentials [9]. However, pairwise functions are included for every constraint which is expensive if many constraints are involved. In contrast, constraints over the feasible instances are considered in [22, 13, 16, 12, 1]. While impressive results have been shown, each different restrictions of the feasible set may require a tailored algorithm. In contrast, we propose to include probabilistic equalities among the model beliefs, which permits derivation of an algorithm that is generally applicable.

## 6    Conclusions

In this work we tackled the problem of inference with belief based equality constraints, which arises when consistency among variables in the network is required. We introduced the CBCBP algorithm that directly incorporates constraints into the CBP framework and results in closed-form update rules. We demonstrated the merit of CBCBP both on synthetic data and on two real-world tasks. Our experiments indicate that CBCBP outperforms PN-potentials in both speed and accuracy. In the future we intend to incorporate our approximate inference with consistency constraints into learning frameworks, *e.g.*, [15, 3].

## Footnotes

[1]Our **distortion limit** $d$ is based on distances from source words to translation slots, rather than distances between source words as in the Moses system [7].

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
