[Reviews · NeurIPS 2016]

Reviewer 1

Summary

The basic idea is that one would like to do convex BP but including certain constraints, e.g. that b(x1)=b(x2). (Note that this is very different from including x1=x2, which is trivial). This is basically done by taking the regular formulation for convex BP where inference is phrased as an optimization over the local Polytope. Then, an extra set of Lagrange multipliers are added to impose the constraint, and the rest of the optimization details go through slightly changed to include these.

Qualitative Assessment

This is an interesting and plausible idea. I only have a few random comments. - On line 63, please describe more clearly the problem with (iv) (Differnce between x1=x2 and b(x1)=b(x2)) - In Eq. 4, should be maximum also take place over v? - I think the general discussion of speed needs more discussion. The introduction implies that it should be faster *per iteration* compared to regular convex BP with a "c penalty". Please formalize how much faster. In addition, the experiments often compare speed, but this is not clearly described enough to be useful. Why is it faster? Is it because of faster iterations or fewer iterations? The experiments don't discuss stopping criteria, basically making them useless as written. (Some discussion of implementation details should also be made.) How do we know the speedup isn't an illusion caused by the convergence threshold? Some plots of, e.g., accuracy vs time would be much more convincing. - In Sections 4.2/4.3, what is the comparison? To some fixed value of c, or to just discarding the constraints?

Confidence in this Review

2-Confident (read it all; understood it all reasonably well)


Reviewer 2

Summary

The authors propose belief propagation for discrete graphical models with certain consistency constraints, related to PN-potentials. The contribution lies in deriving closed-form solutions to the belief-propagation operations in this more general model and showing empirically that it is advantageous over naively doing belief-propagation on general factors explicitly modeling the consistency constraints.

Qualitative Assessment

General comments: (i) The authors only solve a new special kind of higher order consistency constraints, generalizing soft PN-potentials, but not a truly general class of constraints, as indicated in the title or in the abstract. (ii) I do not agree with what the authors say in lines 72 - 78. In case of MAP-inference, which is normally desired, the goal is to obtain a single assignment which satisfies all given linear constraints. The proposed model (i.e. computing marginals) is then less desirable. The relaxed model the authors optimize is simply a byproduct of looking for marginals instead of MAP-assignments (the added entropy is responsible for this). In case of vanishing entropy one gets the same model. Hence there certainly remains the disadvantage of a parameter in the PN-potential, but now hidden in the entropy. Additionally, when a MAP-solution is wanted, the proposed algorithm is in fact disadvantageous, as inference is not done w.r.t. the energy of a single MAP-solution and rounding results in some arbitrariness of the obtained solution. (iii) Experimental comparison: - No comparison against solving MAP-inference with PN-potentials and existing inference algorithms is given. - Experiments are very small scale, e.g. image segmentation is only done on superpixels. I do not consider such microbenchmarks very informative. One can usually just plug everything into an off-the-shelf LP-solver in such cases. - To be able to really judge the algorithm, convergence plots would be helpful, but none are given. In general, I deem experimental comparison unconvincing: while many experiments have been performed, no real comparison against any algorithm other than CBP is performed. (iv) Related work: Many references to work on inference with higher order potentials are lacking. To name a few: - Komodakis: Beyond pairwise energies: Efficient optimization for higher-order MRFs - Tarlow et al.: HOP-MAP: Efficient Message Passing with High Order Potentials. - Kappes: Higher-order Segmentation via Multicuts E.g., the work of Kappes shows how to very efficiently include PN-potentials in an LP-relaxation. Generally, efficient ways for PN-potentials have been explored before, which is not acknowledged in the author's text. (v) The work is rather incremental: The main contribution is the derivation of a new updating formula for convex belief propagation and consistency constraints. Detailed comments: - The entropy H is never defined. - Notation is scattered around: Some is defined in section 2 and some is defined in section 3, but only informally in the text, making the article harder to read. - Line 8: What is the standard approach? - Line 71: which is expect -> which is expected - Line 96: One can also derive duals when the primal is not strictly convex (duals exist even for non-convex programs).

Confidence in this Review

2-Confident (read it all; understood it all reasonably well)


Reviewer 3

Summary

This paper studies adding constraints on the values of beliefs during convexified belief propagation, as opposed to including factors in the model that penalize disagreement between random variables. This is a weaker form of constraint, but the authors argue that it is sufficient for many applications and show that it is less computationally expensive on synthetic data and benchmark problems without sacrificing significant accuracy.

Qualitative Assessment

The proposed method is appealing in that it provides a convenient way to bypass tuning the weights of PN-potentials for convexified BP. My opinion is that it is a straightforward derivation, but one that is worth sharing with the community. The experimental validation is sound. It supports the argument that constrained beliefs are sufficient for enforcing the domain knowledge typically encoded as PN-potentials. If anything, I think the argument that needs to be supported more is the claim that such constraints are useful in practice. It is sometimes considered conventional wisdom, and reinforced by the papers addressing the topic, but it would be good to also evaluate in this manuscript how accurate a model is on these tasks that just includes untuned PN-potentials (so that they're probabilistic dependencies, not hard constraints), as well as some baseline that doesn't include them at all. I'm curious to know how much the inclusion of constraints improves performance. Regarding related work, the discussion on lines 271-276 should be expanded. Some of the cited references, such as [1, 12, 13], use the alternating direction method of multipliers. For this reason, incorporating additional linear constraints that constrain beliefs (or continuous random variables in the case of [1]) is trivial. I think the distinguishing feature of this work is that it incorporates constrained beliefs into a BP algorithm that optimizes the objective using block-coordinate descent. Also, the references [1, 12, 13] are primarily focused on MAP inference. Would there be any complication in extending CBCBP to a max-product version? After author response: I think the authors' plan of including empirical evidence supporting the argument for using constraints will strengthen the manuscript.

Confidence in this Review

3-Expert (read the paper in detail, know the area, quite certain of my opinion)


Reviewer 4

Summary

In this paper, the authors proposed a dual belief propagation algorithm for marginal inference with a special type of constraint --- some variables' assignment have to agree. They enforce this by letting the beliefs of these variables to equal, and add to the local marginal polytope to form a new constraint set. Together with the original linear objective function (with a entropy term), it forms a LP problem, whose dual admits a belief propagation algorithm.

Qualitative Assessment

The idea is straightforward, and no surprise. Minor issues: (1) the title is not informative. It doesn't really tell what the task is and how to do it. (2) 'consistency structure' is not a widely accepted term, perhaps should not be used in the abstract. It would be even better if avoid. (3) The constraint that the proposed BP handles is too special, which restricts its impact. Overall, I am leaning towards accepting it for it's a useful addition to the community. ==Post rebuttal == I still would like to accept it.

Confidence in this Review

3-Expert (read the paper in detail, know the area, quite certain of my opinion)


Reviewer 5

Summary

This paper considers a special case of the approximate marginal inference of discrete Markov random fields, where hard equality constraints of label assignment are enforced for certain nodes. Unlike previous approach [9] who introduces additional consistency potentials, the proposed method explicitly imposes equality constraints to the variational formulation of the marginal inference and solves it with dual coordinate descent. The experiments on synthetic data, image segmentation and machine translation show its computational and statistical efficiency over the baseline [9].

Qualitative Assessment

Technical quality: > The main contribution of this paper is the idea of writing the transitive equality constraints of consistency as simpler equality constraints of common constants. However, I am afraid the transformation (i.e., (4)) is just a relaxation of the original problem rather than an equivalent formulation. One simple argument is that v_k can take different values in [0,1] leading to different optimization problems. The optimal one among those problems gives rise to the original problem. I am also not sure if v_k can be canceled out in the dual problem. The explanation in line 115-116 is vague. Even if we cancel out the linear term, the log-sum-exp function itself could be unbounded below. The soundness of the derived sum-to-zero constraint of nu variables is thus arguable. Also, it seems nontrivial to recover (4) from the dual of the dual given in Lemma 3.1. But all of these could be the matter of details. > Regarding experiments, did you check the beliefs numerically to verify the correctness of those consistency constraints? > It is not 100% clear why sums are taking in (6) without including k and x_r^k, i.e., \sum_{k^\prime \in K_r \setminus k} and \sum_{x_r \setminus x_r^k}. More steps or explanations are needed. Novelty and potential impact: As far as I know, the formulation of label consistency constraint is novel. However, the proposed method works only for a particular type of constraint. Its usage could be limited. Clarity and presentation: This is a well written paper with clear presentation and sufficient experiments. It could be better if the convergence analysis and/or its variants are presented. After rebuttal: The "maximizing over v_k" issue doesn't affect the main results of the paper. However, the proof of lemma 3.1 could be majorly revised: By taking derivative of the Lagrangian w.r.t. v_k, we get the sum-to-zero constraints immediately. Other issues have clear response in the rebuttal.

Confidence in this Review

3-Expert (read the paper in detail, know the area, quite certain of my opinion)


Reviewer 6

Summary

Authors of this paper extend the previous work on convex belief propagation by including consistency constraints. Presented benefits of the proposed model include increased flexibility in constraint definition and computational speedup. Algorithm and proof of convergence is given. Model is compared with the baseline on one synthetic and two real tasks, semantic image segmentation and machine translation.

Qualitative Assessment

This paper introduces an incremental improvement over the previous work on convex belief propagation. Contribution seems minor, but results show notable improvement over the baseline. Overall technical quality of the paper is good, but some parts of the paper could use additional clarification. For example, eq. (1) is hard to follow if the reader is not familiar with prior work. Additional explanation (even a single additional sentence) regarding the constraints would make it much more apprehensible.

Confidence in this Review

2-Confident (read it all; understood it all reasonably well)